# Debias NLU Datasets via Training-free Perturbations

**Qi Guo♠ , Yuanhang Tang♠ , Yawen Ouyang♦ , Zhen Wu♠†, Xinyu Dai♠**
♠National Key Laboratory for Novel Software Technology, Nanjing University
♦Institute for AI Industry Research (AIR), Tsinghua University
{qguo,tangyuanhang}@smail.nju.edu.cn   ouyangyawen@air.tsinghua.edu.cn
{wuz, daixinyu}@nju.edu.cn

## Abstract

Several recent studies have shown that advanced models for natural language understanding (NLU) are prone to capture biased features that are independent of the task but spuriously correlated to labels. Such models often perform well on in-distribution (ID) datasets but fail to generalize to out-of-distribution (OOD) datasets. Existing solutions can be separated into two orthogonal approaches: model-centric methods and data-centric methods. Model-centric methods improve OOD performance at the expense of ID performance. Data-centric strategies usually boost both of them via data-level manipulations such as generative data augmentation. However, the high cost of fine-tuning a generator to produce valid samples limits the potential of such approaches. To address this issue, we propose PDD, a framework that conducts training-free **P**erturbations on samples containing biased features to **D**ebias NLU **D**atasets. PDD works by iteratively conducting perturbations via pre-trained mask language models (MLM). PDD exhibits the advantage of low cost by adopting a training-free perturbation strategy and further improves the label consistency by utilizing label information during perturbations. Extensive experiments demonstrate that PDD shows competitive performance with previous state-of-the-art debiasing strategies. When combined with the model-centric debiasing methods, PDD establishes a new state-of-the-art.

## 1 Introduction

Although recent language models have demonstrated impressive performance on many natural language understanding (NLU) benchmarks (Wang et al., 2018), several studies show that models tend to leverage dataset biases for inference (Poliak et al., 2018; Zhang et al., 2019). Such biases

---

†Corresponding author.

are commonly characterized as spurious correlations between task-independent features and labels (Gardner et al., 2021). These task-independent features exhibiting such spurious correlations are viewed as *biased features*. Models relying on these biased features for inference usually produce superior in-distribution (ID) performance but fail to generalize to out-of-distribution (OOD) datasets. For instance, in the MultiNLI (MNLI) dataset (Williams et al., 2018), negation words are highly correlated with the contradiction label and considered as biased features (Joshi et al., 2022; Shah et al., 2020). Models trained on the MNLI dataset tend to arbitrarily classify pairs containing negation words into the contradiction category.

Previous solutions to resolve the issue can be roughly separated into two types: model-centric methods and data-centric methods (Wu et al., 2022). Model-centric methods boost the robustness of models by designing new training objectives or model architectures to force the models into paying less attention to the biases during training, at the expense of ID performance (Karimi Mahabadi et al., 2020; Clark et al., 2019). Data-centric methods conduct data-level manipulations to debias the original datasets (Ross et al., 2022; Wu et al., 2022; Yang et al., 2020). Recently, data-centric debiasing methods, especially those through generative augmentations, are gaining increasing attention due to their ability to enhance both ID and OOD performance simultaneously.

However, existing data-centric methods based on generative augmentations are still faced with the limitation of high cost (Ross et al., 2022). These methods usually take effort to finetune a large task-specific generator, such as a GPT-2 (Radford et al., 2019), on large-scale NLU datasets to generate new samples. Additionally, they inevitably require costly retraining once the dataset varies, further increasing the computational costs.

To address the limitation of existing data-centric debiasing methods, we propose PDD (**P**erturbing samples containing biased features to **D**ebias NLU **D**atasets), a novel and cost-effective debiasing framework for NLU datasets based on a training-free perturbation strategy. To reduce expenses, we adopt a pre-trained mask language model (MLM) to perturb samples rather than finetuning a generator to generate new samples. Specifically, we perturb the samples containing biased features to mitigate the strong statistical correlations between such features and the labels, which is measured by z-statistics (Gardner et al., 2021). Moreover, we design label-specific prompts to incorporate label information into perturbations and further adopt a confidence filter step to improve label consistency.

To verify the effectiveness of PDD, we carry out experiments on three NLU tasks: natural language inference (Bowman et al., 2015), fact verification (DAGAN et al., 2009), and paraphrase identification. Experimental results demonstrate that PDD shows comparable OOD performance to previous state-of-the-art debiasing methods. Since PDD is data-centric, we enhance it with model-centric methods and observe further improvements in performance, exceeding any previous state-of-the-art to the best of our knowledge. Our code and data are available at: https://github.com/GuoQi2000/Debias_PDD.

In summary, our contributions include:

- We introduce PDD, an effective and low-cost dataset debiasing framework based on a training-free perturbation strategy, compensating for the drawbacks of existing data-centric debiasing methods.

- We conduct extensive experiments on several NLU benchmarks and demonstrate that our framework outperforms strong baselines on several ODD datasets.

## 2 Related Work

**Biases in Datasets:** Biases in NLU benchmarks are idiosyncratic artifacts introduced in the annotation processes and are often modeled as spurious correlations between simple features and labels. Several works trying to provide a theoretical framework to measure such spurious correlations. Gururangan et al. (2018) involves pointwise mutual information (PMI) to quantify the spurious correlations in datasets. Veitch

et al. (2021) formalize spurious correlations in a causal framework and consider the model's prediction should be invariant to perturbations of the spurious feature. Gardner et al. (2021) assume a uniform distribution obeyed by the prediction conditioned on any single feature and figure out spurious features with hypothesis testing.

**Model-centric Methods:** Model-centric strategies aim to prevent models from overfitting the biased features in samples during training. Belinkov et al. (2019b) improves the robustness of models on OOD datasets through the removal of biased features at a representative level with adversarial training techniques. Product-of-Expert (Karimi Mahabadi et al., 2020) and its variant Learned-Mixin (Clark et al., 2019) debias the model by adjusting the loss function to downweight the samples that can be easily solved by bias-only models. Tang et al. (2023) capture the model biases automatically by shuffling the words of the input sample and further debias the models in product of experts. Utama et al. (2020) introduce a confidence regularization method to discourage models from exploiting biases, without harming the ID performance. Yaghoobzadeh et al. (2021) robustify models by fine-tuning the models twice, first on the full data and second on the minorities with low confidence.

**Data-Centric Methods:** Several works try to improve the robustness by conducting data-level manipulations such as data augmentations and data filtering. Bartolo et al. (2021) generate adversarial datasets to improve the robustness of question answering models. Le Bras et al. (2020) propose AFLITE which adjusts the distribution through adversarial filtering. Wu et al. (2022) fine-tunes a GPT-2 (Radford et al., 2019) to fit the original dataset, then conducts spurious filtering to generate a debiased dataset. Ross et al. (2022) alleviate the syntactic biases in datasets by semantically-controlled perturbations to generate samples with high semantic diversity. Different from these approaches, PDD generates samples via a training-free perturbation strategy and can be applied to a variety of known dataset biases.

## 3 Methodology

In this section, we describe how our framework PDD works in detail. We start by selecting a set

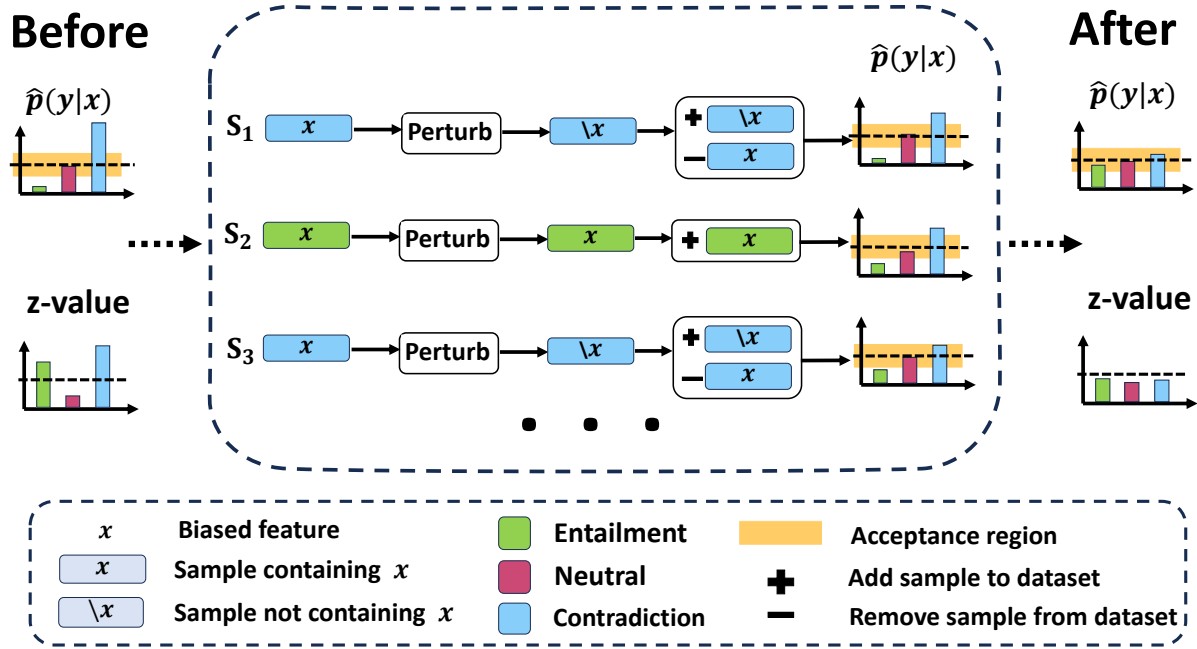

Figure 1: An overview of PDD. The z-value of $x$ is rejected by the uniform hypothesis (above the threshold). To reduce its z-value, we iterate through the samples that contain $x$ and conduct perturbations to generate new samples to make $\hat{p}$ closer to $p_0$. Each time a sample is perturbed, we update the $\hat{p}$ and z-value. The process is stopped when the z-values over all labels are under the threshold.

of task-independent features based on prior knowledge (Poliak et al., 2018; McCoy et al., 2019) and quantify their correlations to the labels via *z-value* so as to identify the biased features (section 3.1). Then we iteratively conduct perturbations on the samples containing biased features to reduce the z-value and produce debiased datasets (section 3.2). The key component — a training-free perturbation strategy — is described separately (section 3.3).

Throughout this section, we use the NLI task as an example for illustration. Note that our framework is not limited to the NLI task but is applicable to a wide range of NLU tasks.

### 3.1 Quantifying Dataset Biases

We start by selecting a set of task-independent features that are widely studied in previous works (Gururangan et al., 2018; McCoy et al., 2019; Poliak et al., 2018). Specifically, we select the following three features: (1) *unigrams*; (2) *lexical overlap* (the number of words shared by premise and hypothesis normalized by the length of hypothesis); (3) *hypothesis-only predictions* (the predictions of a model that takes only the hypothesis as the input).

To identify biased features and quantify the dataset biases, we follow Gardner et al. (2021)

to utilize a hypothesis testing technique. The assumption behind is that the posterior probability given only a task-independent feature should follow a uniform distribution. And the features violating the assumption are considered as biased features. More formally, for each task-independent feature $x$, let $N$ be the number of samples containing $x$, $K$ the number of classes, and $p_0 = 1/K$ the ideal probability following the uniform distribution. The conditional probability $p(y = l|x)$, which is approximated by its expectation value $\hat{p}(y = l|x) = \frac{1}{N}\Sigma_{i=1}^{N}\mathbb{I}(y_i = l)$, should be uniform over the class labels $l$. Under the uniform hypothesis: $\hat{p}(y = l|x) = p_0$, we compute the standardized z-statistics, which measures the deviation of $\hat{p}$ from the uniform distribution:

$$z^*(l, x) = \frac{\hat{p} - p_0}{\sqrt{p_0(1 - p_0)/n}}. \tag{1}$$

We refer to the absolute value of *z-statistic* as *z-value*. The larger the z-value is, the stronger the correlation between $x$ and $l$. We consider the features whose z-value exceeds a predefined threshold $\tau$ as the biased features and collect them as a biased feature set $X_b$.

## 3.2 Debiasing Datasets by Reducing Z-value

Under the uniform hypothesis, debiasing a dataset is equivalent to weakening the correlation between the task-independent features and the labels until their z-values are all within the threshold. To achieve this, we iteratively perturb the samples that contain biased features to reduce the z-value.

Let $D_b$ denote the original biased dataset, and $D$ the debiased dataset. we initialize the debiased dataset $D = D_b$. For each $x \in X_b$, we compute the z-values of $x$ over all labels. Then we iterate through the samples $(P, H, l)$ that contain $x$ and conduct perturbations on those samples with z-value exceeding the threshold $\tau$. Those samples to be perturbed can be separated into two cases: (1) $\hat{p}(y = l|x) < p_0$ and (2) $\hat{p}(y = l|x) > p_0$. For each sample in the first case where $\hat{p}(y = l|x) < p_0$, we perturb it to generate a new sample $S' = (P', H', l)$ still **containing** $x$ and add it to $D$, so as to increase $\hat{p}$. For each sample in the second case where $\hat{p}(y = l|x) > p_0$, we perturb it to get $S' = (P', H', l)$ **not containing** $x$ and add it to $D$ and remove the original sample $S$ from $D$, so as to decrease $\hat{p}$. Both the two operations above aim to reduce the z-value by bringing the $\hat{p}(y = l|x)$ closer to $p_0$ (see appendix C for proof). The perturbations are conducted iteratively until the z-values of all $x$ over all labels are under the threshold $\tau$. The perturbation strategies are detailed in the next section and the full algorithm is demonstrated in Figure 1 and Algorithm 1.

## 3.3 Training-free Perturbations

In order to perturb the samples to generate new samples at a low cost, we abandon the previous way to fine-tune a large generator and instead propose a training-free perturbation strategy consisting of two stages: (1) feature-specific masking and (2) label-preserving augmentation via MLM. The first stage regulates the appearance of biased features in the generated samples by masking tokens and the second stage helps generate samples while preserving the original labels. The whole procedure is depicted in Figure 2.

### 3.3.1 Feature-specific Masking

Since the biased features vary, we manually design different masking rules for different biased features according to the value of $\hat{p}$ (listed in appendix A.1).

• $\hat{p}(y = l|x) < p_0$ : In this case, we generate new samples still containing $x$ to increase $\hat{p}(y = l|x)$ so as to bring it closer to $p_0$. Therefore,

---

**Algorithm 1:** Z-value Reducing

**Input:** Biased dataset $D_b$; Biased feature set $X_b$; Number of labels $K$; Threshold $\tau$
**Output:** Debiased dataset $D$.

Initialize $D \leftarrow D_b$.
**for** $x \in X_b$ **do**
    Compute $z^*(l, x)$ $(l = 1, ..., K)$
    **while** $\max(|z^*(l, x)|) > \tau$ **do**
        **for** $S = (P, H, l) \in D_b$ **do**
            **if** $S$ contains $x$ and $|z^*(l, x)| > \tau$ **then**
                **if** $\hat{p}(y = l|x) < p_0$ **then**
                    Perturb $S$ to get
                      $S' = (P', H', l)$
                    **containing** $x$
                  Add $S'$ to $D$
                **end**
                **else**
                  Perturb $S$ to get
                    $S' = (P', H', l)$
                  **not containing** $x$
                Add $S'$ to $D$
                Remove $S$ from $D$
                **end**
             Update $z^*(l, x)$
            **end**
        **end**
    **end**
    $D_b \leftarrow D$
**end**

---

$x$ should be retained when masking words. To be specific, for unigrams, we skip the biased words and randomly mask the rest of the words to a fixed percentage. For lexical overlap, we retain the overlapped words in the premise and the hypothesis and randomly mask the rest. For hypothesis-only predictions, we retain the hypothesis and randomly mask the premise.

• $\hat{p}(y = l|x) > p_0$ : In this case, we generate new samples not containing $x$ to decrease $\hat{p}(y = l|x)$ to bring it closer to $p_0$. Therefore, $x$ should be eliminated in the masking stage. Specifically, for unigrams, we mask the biased word and the rest randomly to a fixed percentage. For lexical overlap, we mask the overlapped words and the rest to a fixed percentage. In terms of the hypothesis-only predictions, we randomly mask the hypothesis.

### 3.3.2 Label-preserving Augmentation via MLM

After masking, we employ MLM to fill in the blanks to generate new samples. To improve label consistency, we propose prompt-based filling to encourage the MLM to preserve the labels and a confidence filtering step to filter out the generated samples with inconsistent labels.

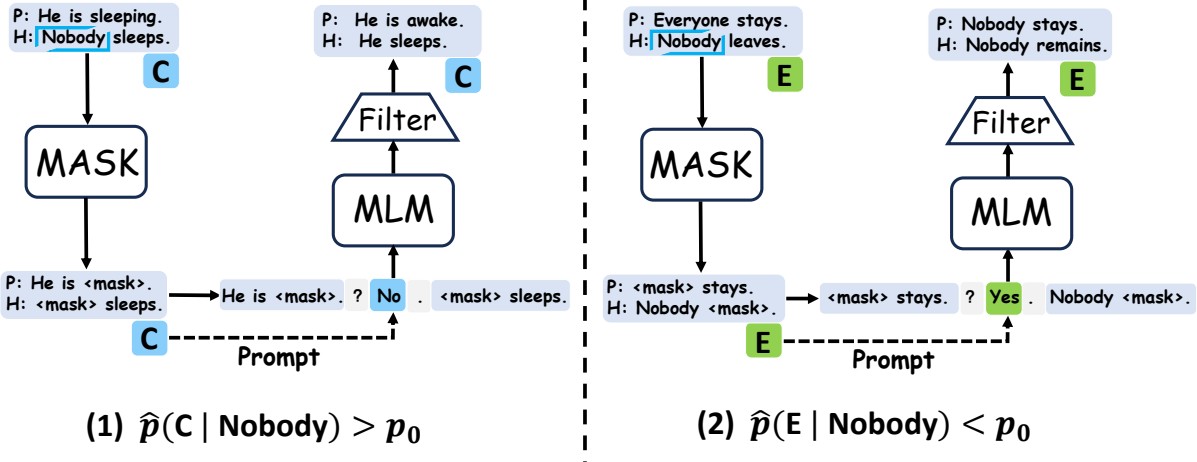

Figure 2: An illustration of our perturbations in NLI. Each sample consists of a premise (P) and a hypothesis (H). E, N, and C represent entailment, neutral, and contradiction respectively. The biased word "Nobody" is highly correlated with the label and rejected by the uniform hypothesis testing. In the case $\hat{p} > p_0$ (left), we mask "nobody" and adopt prompt-based filling to generate a batch of candidates. Then we adopt a confidence filter to select the one with the highest confidence on the correct label. In the case $\hat{p} < p_0$ (right), we preserve "nobody" during masking, followed by a similar subsequent augmentation process.

**Prompt-Based Filling:** To improve label consistency of generated samples, we design label-specific prompts to incorporate the label information into the input of the MLM. In this way, the MLM is encouraged to take the label information into consideration. To build such prompts, we follow Zhou et al. (2022) to map the label $l$ to a relevant prompting word $[w]$. Specifically, We map "entailment" to "Yes", "neutral" to "Maybe", and "contradiction" to "No". We feed the connected input "$P_{mask}?[w].H_{mask}$" into the MLM, where $P_{mask}$ denotes the masked premise and $H_{mask}$ denotes the masked hypothesis, to encourage it to generate samples with correct labels.

**Confidence Filtering:** Though label-specific prompts help to preserve the labels, the MLM may still generate samples that are inconsistent with the labels. To address this, we follow previous works (Yang et al., 2020; Wu et al., 2022) to train a BERT-base model on the original dataset and use it as a filter. For each masked sample, we generate various candidates and feed them to the filter. We choose the one with the highest output confidence to maintain label consistency.

## 4 Experimental Setup

To validate the effectiveness of PDD, we conduct experiments on three NLU tasks: natural language

inference, fact verification, and paraphrase identification. We compare PDD with existing state-of-the-art debiasing strategies on several OOD challenging datasets (Karimi Mahabadi et al., 2020; Clark et al., 2019; Utama et al., 2020; Meissner et al., 2022; Wu et al., 2022; Sanh et al., 2021).

### 4.1 Natural Language Inference

**Source Dataset:** Following previous works (Poliak et al., 2018; McCoy et al., 2019), we use the MultiNLI (MNLI) (Williams et al., 2018) as the original dataset. MNLI has two ID test sets (MNLI-m and MNLI-mm), one that matches the domains of the training data and one with mismatching domains.

**Biases in Dataset:** (1) syntactic bias: McCoy et al. (2019) find that models tend to exploit syntactic biases in datasets (e.g., the syntactic overlapping between the premise and hypothesis is strongly correlated with the entailment label). (2) hypothesis-only bias: Poliak et al. (2018) shows that a hypothesis-only model that can only see the hypothesis during training captures statistical cues in the hypothesis and succeed in predicting the majority of the test samples accurately.

**Debiased Dataset:** Considering the biases in MNLI, we select three features: (1) unigrams;

(2) lexical overlap > 0.8; (3) hypothesis-only predictions and debias them one by one. We first debias the hypothesis-only prediction, then the lexical overlap, and finally the unigrams. The order is determined by ranking the z-value from highest to lowest. Details of our debiased datasets are shown in appendix A.

**OOD Evaluation Dataset:** For syntactic bias, we adopt HANS (McCoy et al., 2019), a challenging evaluation dataset where the lexical heuristics in MNLI fails. For hypothesis-only bias, we choose the MNLI-hard (Belinkov et al., 2019a) that is constructed from the two ID test datasets by filtering out the samples that can be accurately predicted by a hypothesis-only model.

## 4.2 Fact Verification

**Source Dataset:** This task involves verifying a claim sentence in the context of a given evidence sentence. We use the FEVER dataset (Aly et al., 2021) that consists of pairs of claims and evidence generated from Wikipedia. FEVER consists of samples with three labels: support, refutes, and not enough information.

**Biases in Dataset:** Claim-only bias: Schuster et al. (2019) find that in FEVER the occurrence of words and phrases in the claim is biased toward certain labels, which is exploited by claim-only models for inference.

**Debiased Dataset:** We select two features (1) unigrams and (2) claim-only predictions, and debias the claim-only predictions first then the unigrams. The order is determined by ranking the z-value from highest to lowest. Note that samples with "not enough information" labels just account for a fairly small proportion of the whole training set, which makes it difficult to reduce the z-value. To address the difficulty, we view "refutes" and "not enough information" as one class when computing z-values.

**OOD Evaluation Dataset:** For claim-only bias, we use the Fever-symmetric dataset (Schuster et al., 2019) for evaluation. This challenging set is constructed by making claim and evidence appear with each label to avoid the idiosyncrasies in claims. Hence, models relying on biases in claims fail on the challenging OOD datasets.

## 4.3 Paraphrase Identification

**Source Dataset:** This task is to determine whether two given questions are semantically matching (duplicate) or not. We adopt the Quora Question Pairs (QQP) (DataCanary, 2017) dataset for debiasing. QQP consists of pairs of questions which are labeled duplicate or non-duplicate.

**Biases in Dataset:** Syntactic bias: In QQP, pairs with low word overlap (<0.8) are heavily unbalanced over labels, indicating a severe lexical overlap bias in QQP. Models trained on QQP heavily rely on lexical-overlap features for inference and fail on negative question pairs with high word overlap (Zhang et al., 2019).

**Debiased Dataset:** We consider two features: (1) lexical overlap < 0.8 and (2) unigrams, and debias them in descending order of z-value. Notice that samples in QQP with low lexical overlap are heavily biased to the non-duplicate label. Hence we choose lexical overlap < 0.8 instead of lexical overlap > 0.8.

**OOD Evaluation Dataset:** We select the QQP subset of PAWS (Paraphrase Adversaries from Word Scrambling) (Zhang et al., 2019) for evaluation. Paws contains 108,463 pairs where the majority of non-paraphrase pairs have high lexical overlap. Models trained on QQP fail miserably on PAWS, much worse than the random baseline.

## 4.4 Compared Methods

For a comprehensive evaluation, we compare PDD against a range of existing debiasing methods.

- **BERT-base (baseline)** (Devlin et al., 2019) which shows impressive performance on different NLU benchmarks (Wang et al., 2018), is considered as the base model.

- **Product-of-Expert** (**PoE**) (Karimi Mahabadi et al., 2020) ensembles a bias-only prediction with the prediction of the main model to enforce the main model to focus on the samples that are not predicted correctly by the bias-only model.

- **Learned-Mixin** (Clark et al., 2019) improves the performance by proposing a learnable weight for the bias-only model's predictions.

| Method | MNLI-m | | | | MNLI-mm | | | |
|---|---|---|---|---|---|---|---|---|
| | ID | | OOD | | ID | | OOD | |
| | dev | test | dev-hard | test-hard | dev | test | dev-hard | test-hard |
| BERT-base (baseline) | 84.06 | 84.69 | 76.42 | 76.21 | 84.65 | 83.49 | 77.71 | 75.68 |
| PoE (Karimi Mahabadi et al., 2020)♠* | 84.6 | 84.1 | 78.2 | 77.5 | 84.9 | 83.5 | 79.1 | 78.3 |
| Learned-Mixin (Clark et al., 2019)♠* | 80.5 | 79.5 | - | **79.2** | 81.2 | 80.4 | - | 78.2 |
| Reg-conf (Utama et al., 2020)♠* | 84.6 | 84.1 | - | 78.3 | 85.0 | **84.2** | - | 77.3 |
| Z-aug (Wu et al., 2022)♦* | 84.72 | **85.12** | 78.95 | 78.60 | **85.14** | 84.09 | **80.29** | **78.51** |
| PDD (ours)♦ | **84.81** | 85.09 | **79.12** | 78.67 | 84.95 | 83.73 | 80.06 | 78.27 |
| PoE + MNLI (baseline) | 81.84 | 82.32 | 78.19 | 78.23 | 82.31 | 81.29 | 79.58 | 77.01 |
| PoE + Z-aug | 82.39 | 83.47 | 79.86 | **79.84** | 82.35 | 82.15 | **81.12** | 78.92 |
| PoE + PDD (ours) | **83.14** | **83.72** | **79.88** | 79.79 | **83.28** | **82.42** | 81.06 | **78.95** |

Table 1: Accuracy of models on the in-distribution datasets of MNLI-m and MNLI-mm along with their hard datasets: MNLI-m hard and MNLI-mm hard. We present the best results in **bold** and the second with underlines. ♠ and ♦ represent the model-centric methods and data-centric methods respectively. * are results reported in previous works.

- **Reg-conf** (Utama et al., 2020) utilizes confidence regularisation to avoid the drop in ID performance and simultaneously improves the OOD performance.

- **Reweighting** (Schuster et al., 2019) assigns a balancing weight to each training sample based on the correlation of the n-grams within the claims with labels for FEVER.

- **Debiasing-Masking** (Meissner et al., 2022) assumes that bias in NLI is caused by a certain subset of weights in the network and designs a mask search to remove the biased weights.

- **Weak-Learner** (Sanh et al., 2021) leverages limited capacity models which learn to exploit biases to train a more robust model via PoE.

- **Z-aug** (Wu et al., 2022), a typical data-centric debiasing method for NLI datasets, augments samples via a fine-tuned GPT-2 with a subsequent z-filtering step.

**Enhancing PDD with Model-Centric Methods:** Since PDD is a data-centric debiasing method and is agnostic to the models, we examine whether model-centric debiasing methods can also benefit from our debiased datasets and further improve the robustness of models. Following (Wu et al., 2022), we experiment PoE on our debiased MNLI dataset and evaluate on MNLI and HANS (implementation details listed in appendix D).

### 4.5 Implementation

We adopt the BERT-base model (Devlin et al., 2019) as the backbone for all the methods. As for PDD, we adopt XLM-large (Conneau et al., 2019) as the MLM and train the BERT-base model on the debiased dataset produced by PDD. Experiments are repeated 5 times with different random seeds and the average scores are reported. More implementation details are listed in appendix B.

## 5 Results

Experimental results on MNLI-hard, HANS, FEVER and QQP are shown in Table 1, Table 2, Table 3, and Table 4 respectively. The main results are summarized as follows.

- **PDD outperforms the BERT baseline on both ID and OOD datasets.** Compared with the BERT-base model trained on the original dataset, PDD greatly boosts the OOD accuracy by 5.71% on HANS, 2.46% on MNLI-m hard, 2.59% on MNLI-mm, 4.43% on FEVER-symmetric and 6.49% on PAWS respectively. Additionally, PDD also shows better ID performance than the baseline on these benchmarks except QQP. The substantial ID improvements on FEVER (85.66→88.51) may be attributed to the distributional gap between the training set and the development set.

- **PDD demonstrates competitive performance with existing state-of-the-art methods.** On MNLI-hard and HANS, PDD obtains similar improvements to model-centric debiasing methods. On FEVER, we notice that PDD achieves the best scores on both the development set (88.51) and the symmetric set (62.48).

- **PDD yields further improvements when combined with model-centric methods.** After conducting PoE on our debiased dataset, we find that models exceed any single debiasing method

and establish a new state-of-the-art on HANS to the best of our knowledge.

| Method | HANS |
|---|---|
| BERT-base (baseline) | 63.25 |
| PoE(Karimi Mahabadi et al., 2020)♠* | 66.31 |
| Learned-Mixin (Clark et al., 2019)♠* | 66.15 |
| Reg-conf (Utama et al., 2020)♠* | **69.1** |
| Reg-conf$_{self-debias}$ (Utama et al., 2020)♠* | 67.1 |
| Weak-Learner (Sanh et al., 2021)♠* | 67.9 |
| Debiasing-Masking (Meissner et al., 2022)♠* | 68.69 |
| Z-aug (Wu et al., 2022)♦* | 67.69 |
| PDD (ours) ♦ | 68.96 |
| PoE+ MNLI (baseline) | 68.84 |
| PoE + Z-aug | 72.04 |
| PoE + PDD (ours) | **72.86** |

Table 2: The accuracy of models evaluated on HANS.

| Method | dev | symmetric |
|---|---|---|
| BERT-base (baseline) | 85.66 | 58.05 |
| PoE♠ (Karimi Mahabadi et al., 2020)* | 83.3 | 62.0 |
| Learned-Mixin♠ (Clark et al., 2019)* | 83.1 | 60.4 |
| Reg-conf♠ (Utama et al., 2020)* | 86.4 | 60.5 |
| Reg-conf$_{self-debias}$♠ (Utama et al., 2020)* | 87.6 | 60.2 |
| Reweighting♠ (Schuster et al., 2019)* | 85.5 | 61.7 |
| Weak-Learner♠ (Sanh et al., 2021)* | 86.4 | 58.5 |
| PDD♦ (ours) | **88.51** | **62.48** |

Table 3: The accuracy of the models on the development set of FEVER and FEVER-symmetric.

| Method | dev | PAWS |
|---|---|---|
| BERT-base (baseline) | **91.27** | 35.82 |
| PoE♠ (Karimi Mahabadi et al., 2020)* | 86.9 | **56.5** |
| Learned-Mixin♠ (Clark et al., 2019)* | 87.6 | 55.7 |
| Reg-conf♠ (Utama et al., 2020)* | 89.1 | 40.0 |
| Reg-conf$_{self-debias}$♠ (Utama et al., 2020)* | 89.0 | 43.0 |
| Reweighting♠ (Schuster et al., 2019)* | 89.5 | 48.6 |
| Debiasing-Masking♠ (Meissner et al., 2022)* | 89.6 | 44.3 |
| PDD♦ (ours) | 89.37 | 42.31 |

Table 4: The accuracy of the models on the development set of QQP and PAWS.

# 6 Discussions and Analysis

## 6.1 Generalisation to Larger PLMs

We verify whether PDD is able to generalize to larger and more powerful Pre-trained Language Models (PLMs). We select three widely-used PLMs as substitute for Bert-base model: Roberta-base, Roberta-large (Zhuang et al., 2021), and Alberta-xxlarge (Lan et al., 2019).

As observed in Table 5, **PDD successfully generalizes to larger PLMs**, yielding average gain of

1.79%, 1.38%, 0.59% for Roberta-base, Roberta-large and Albert-xxlarge respectively. Additionally, great improvements in accuracy are observed on the HANS dataset. One possible reason is that the feature-specific masking helps to relieve the syntactic heuristic by directly masking the words shared between the premise and the hypothesis.

| Model | MNLI | PDD | $\Delta$ |
|---|---|---|---|
| **On MNLI-m hard** | | | |
| Roberta-base | 81.52 | 82.20 | +0.68 |
| Roberta-large | 85.24 | 86.21 | +0.97 |
| Albert-xxlarge | 85.74 | 86.27 | +0.53 |
| **On MNLI-mm hard** | | | |
| Roberta-base | 81.92 | 82.83 | +0.91 |
| Roberta-large | 85.21 | 85.56 | +0.35 |
| Albert-xxlarge | 86.13 | 85.98 | -0.15 |
| **On HANS** | | | |
| Roberta-base | 73.86 | 77.65 | +3.79 |
| Roberta-large | 77.48 | 80.31 | +2.83 |
| Albert-xxlarge | 76.31 | 77.71 | +1.40 |

Table 5: Performance of larger PLMs on MNLI.

## 6.2 Debiasing Augmentation vs Common Data Augmentation

To verify whether the improvement of robustness on OOD datasets results from our perturbations or just from the increment of data size, we contrast our method with two common augmentation strategies: Easy Data Augmentation (EDA) (Wei and Zou, 2019) and Back Translation (BT) (Fadaee et al., 2017; Sennrich et al., 2016; Yu et al., 2018).

Figure 3 shows the results. **The increment of data size does not necessarily lead to better OOD performance and the improvement of robustness mainly results from our perturbations.** PDD achieves higher accuracy on both MNLI-hard and HANS, suggesting the models actually benefit from our perturbations.

While the back translations help improve the OOD accuracy, EDA harms it. Additionally, we exploit the potential of PDD as a regular augmentation framework by setting the threshold of z-value to 0 to produce more samples and finally get a debiased dataset with 740K samples. We find slight performance decrement after augmentation. One possible explanation is that the increment of data size introduces unknown biases.

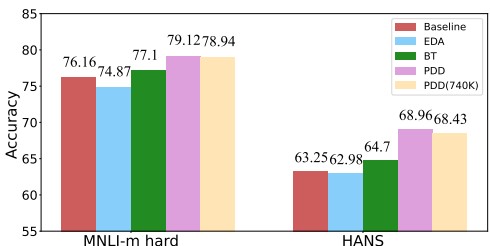

Figure 3: Accuracy of models trained on different augmented datasets on MNLI-m hard (left) and HANS (right).

## 6.3 Transfer Performance

Following Karimi Mahabadi et al. (2020), we evaluate how well PDD can generalize to multiple out-of-domain test sets. We select 7 different datasets including MPE (Lai et al., 2017), MRPC (el Said et al., 2015), SCITAIL (Khot et al., 2018), SICK (Marelli et al., 2014), QNLI, WNLI, and RTE (Wang et al., 2018). We take the Bert-base models trained on debiased MNLI and test them on the development sets of these datasets.

**PDD achieves substantial improvement in generalization to multiple OOD datasets** (Table 6). Models trained on debiased MNLI achieve an average gain of 1.24% on accuracy and perform better than the baseline on 6 of these datasets.

| Dataset | Baseline | PDD | $\Delta$ |
|---|---|---|---|
| MPE | 75.07 | 76.68 | +1.61 |
| MRPC | 57.84 | 59.31 | +1.47 |
| QNLI | 50.61 | 50.86 | +0.25 |
| WNLI | 43.66 | 45.07 | +1.41 |
| SCITAIL | 76.99 | 77.99 | +1.00 |
| SICK | 55.13 | 54.07 | -1.06 |
| RTE | 74.37 | 78.34 | +3.97 |

Table 6: Accuracy of models trained on debiased MNLI transferring to new target datasets.

| Method | MNLI-m hard | HANS |
|---|---|---|
| Baseline | 76.42 | 63.25 |
| Full method | 79.12 | 68.96 |
|    w/o unigrams | $78.56_{-0.56}$ | $67.58_{-1.38}$ |
|    w/o lexical overlap | $78.80_{-0.32}$ | $66.49_{-2.47}$ |
|    w/o hyp-only prediction | $76.70_{-2.42}$ | $64.52_{-4.44}$ |
|    w/o confidence filtering | $74.87_{-4.25}$ | $56.96_{-12.00}$ |

Table 7: Ablation study conducted on MNLI-m hard and HANS.

## 6.4 Ablation Study

We perform ablation studies to assess the impact of different components in our framework including (1) biased features and (2) the confidence filter.

Results on MNLI-m and HANS are presented in Table 7. **All the components contribute to the improvement of model robustness.** A Performance drop is observed when any of the selected components is removed. Even the removal of hypothesis-only prediction which only targets the hypothesis-only bias influences the performance of models on HANS, indicating that those features are not independent and all contribute to the robustness of models.

## 7 Conclusions

We propose PDD, a novel and cost-effective debiasing framework for NLU datasets adopting a training-free perturbation strategy to mitigate the statistical correlations between biased features and labels. In contrast with existing data-centric strategies, PDD shows applicability to various biases in different datasets at a low cost. Extensive experiments demonstrate its effectiveness over existing baselines. When combined with model-centric methods, PDD further improves the OOD performance, achieving a new state-of-the-art. Future work may concern extending PDD with automatic identifications of biased features.

## Limitations

Despite that PDD achieves impressive results on several challenging OOD datasets, it still has the following limitations: (1) The selection of biased features heavily relies on prior knowledge. (2) Since our algorithm to reduce the z-value is a greedy one, the operation to mitigate the z-value of one class may influence the z-values of other classes. It is difficult to balance the z-value over all classes for tasks with a large size of labels.

## Acknowledgements

We would like to thank the anonymous reviewers for their constructive comments. This work was supported by the National Natural Science Foundation of China Projects (No. 62206126, 61936012 and 61976114).

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

# A Debiased datasets

## A.1 Data Size of Debiased Datasets

| Dataset | Original | Debiased |
|---------|----------|----------|
| MNLI | 391,120 | 594,220 |
| FEVER | 242,911 | 284,767 |
| QQP | 384,346 | 571,706 |

Table 8: Data size of our debiased datasets.

## A.2 Debiased MNLI

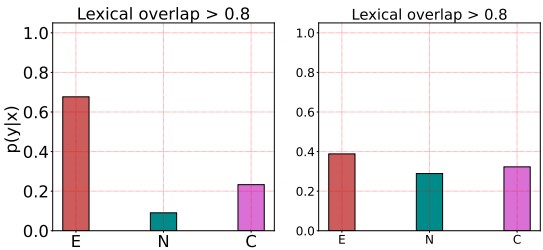

Figure 4: The label distribution of samples with lexical overlap > 0.8 in MNLI (left) and debiased MNLI (right).

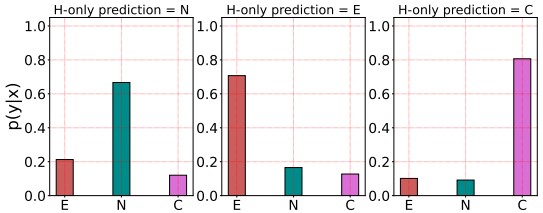

Figure 5: The label distribution of samples under different predictions of hypothesis-only model in MNLI.

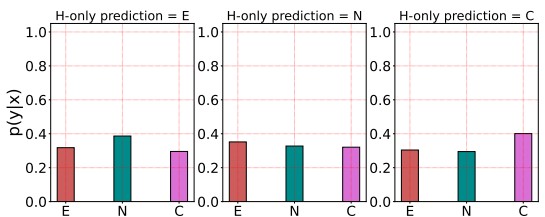

Figure 6: The label distribution of samples under different predictions of hypothesis-only model in debiased MNLI.

## A.3 Debiased FEVER

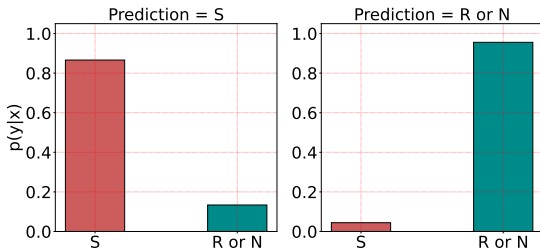

Figure 7: The label distribution of samples under different predictions of claim-only model in FEVER.

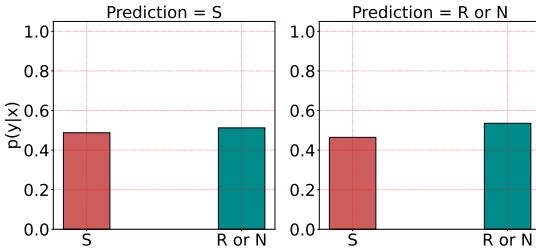

Figure 8: The label distribution of samples under different predictions of claim-only model in debiased FEVER.

## A.4 Debiased QQP

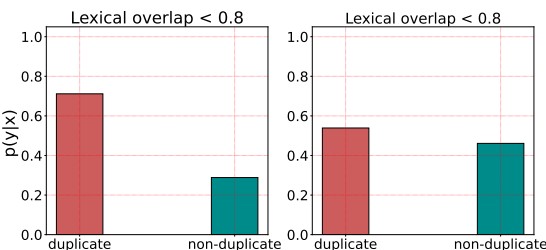

Figure 9: The label distribution of samples with lexical overlap < 0.8 in QQP (left) and debiased QQP (right).

## A.5 Masking Rules

We manually design the masking rules for different biased features under two situations $\hat{p} > p_0$ and $\hat{p} < p_0$. All our masking rules derive from a random masking method and are listed in table 9.

| feature | masking rules | |
|---|---|---|
| | $\hat{p} > p_0$ | $\hat{p} < p_0$ |
| unigrams | mask the single word | mask the rest |
| lexical overlap | mask common words in $P$ and $H$ | mask the rest |
| hypothesis-only prediction | mask words of $H$ | mask words of $P$ |
| claim-only prediction | mask words of $C$ | mask words of $E$ |

Table 9: List of masking rules. P and H denote the premise and hypothesis. C and E denote the claim and evidence. All the masking rules derive from a random masking strategy: we mask the words of the samples to a fixed percentage.

## B   Hyper-Parameters

We carry out all our experiments in a Linux environment with a single RTX 3090 (24G). B.2, B.3 and B.4 list the details of hyperparameters for fine-tuning a BERT-base model. For roberta-base, roberta-large, and Albert, we fix other hyperparameters and just change the batch size (32 for roberta-base, 16 for roberta-large, and 8 for Albert) due to the limitation of GPU memory. We roughly spend 30 hours to debias MNLI and 20 hours for FEVER and QQP.

### B.1   Perturbations

| Hyperparameter | Value |
|---|---|
| Masking ratio for unigram | 0.25 |
| Masking ratio for H-only prediction | 0.4 |
| Masking ratio for lexical overlap | 0.4 |
| Threshold $\tau$ | 20 |
| Pre-trained MLM | XLM-large |

Table 10: Hyperparameter for perturbations

### B.2   NLI Model Training

| Hyperparameter | Value |
|---|---|
| Learning rate | 2e-5 |
| Batch size | 32 |
| Loss | Cross-Entropy |
| Epoch | 5 |
| Optimizer | Adam (default) |
| Learning rate scheduler | Linear |
| Warm up steps | 2000 |
| Max sequence length | 128 |

Table 11: Hyperparameter for training NLI models on MNLI and our debiased MNLI

### B.3   FEVER Model Training

| Hyperparameter | Value |
|---|---|
| Learning rate | 1e-5 |
| Batch size | 8 |
| Loss | Cross-Entropy |
| Epoch | 5 |
| Optimizer | Adam(default) |
| Learning rate scheduler | Linear |
| Warm up steps | 1500 |
| Max sequence length | 128 |

Table 12: Hyperparameter for training models on FEVER and our debiased FEVER

### B.4   QQP Model Training

| Hyperparameter | Value |
|---|---|
| Learning rate | 1e-5 |
| Batch size | 16 |
| Loss | Cross-Entropy |
| Epoch | 5 |
| Optimizer | Adam(default) |
| Learning rate scheduler | Linear |
| Warm up steps | 2000 |
| Max sequence length | 128 |

Table 13: Hyperparameter for training models on QQP and our debiased QQP

## C   Reducing Z-values

Given feature $x$ and label $l$, the object z-value to optimize is

$$|z^*(l, x)| = |\frac{\hat{p} - p_0}{\sqrt{p_0(1 - p_0)/n}}|.$$

We donate the number of samples under each label as $n_1, n_2, ..., n_K$. $n = \Sigma_{i=1}^{K} n_i$ $\hat{p} = \frac{n_l}{n}$

Reducing z-value equals to reduce

$$z^*(l,x)^2 = (\frac{\hat{p} - p_0}{\sqrt{p_0(1-p_0)/n}})^2$$

$$\propto n(\hat{p} - p_0)^2$$

When we conduct the perturbations we can only see a sample with its label $l$, so we just consider partial derivative along $n_l$:

$$\frac{\partial z^*(l,x)^2}{\partial n_l} = \frac{2n_l}{n} - \frac{n_l{}^2}{n^2} - 2p_0 + p_0{}^2$$

$$= (2 + p_0 + \hat{p})(p_0 - \hat{p})$$

Hence the operation on $n_l$ to reduce z-value actually work by minimizing the gap between $\hat{p}$ and $p_0$. For $\hat{p} < p_0$ we increase $\hat{p}$ by $n_l + 1$ and for $\hat{p} < p_0$ we decrease $\hat{p}$ by $n_l - 1$.

## D  Details of Combining PoE with PDD

### D.1  Bias-Only Model for Hypothesis-Only Bias

We consider the BERT-base model that can only see the hypothesis during training to model the artifacts in the hypothesis.

### D.2  Bias-Only Model for Syntactic Bias

Instead of using the manually designed features proposed by (Karimi Mahabadi et al., 2020), we train a BERT-base model that can only see the unordered words to capture the syntactic bias (Tang et al., 2023).

## E  Perturbation Strategies

In Table 14, we illustrate our masking rules for MNLI with concrete examples.

**unigrams**

Original: P: It's kind of strange here the way things go uh here if you have an accident and no one's injured the police won't even show up.
H: The police don't show up if nobody is hurt.
l: entailment

Biased feature: $x = $ nobody    $\hat{p}(l|x) < \frac{1}{3}$
Masking rule: Preserve nobody, randomly mask other words.
Masked: P: It's kind of <mask> here the way we go uh here if you have an accident and no one's <mask> the police won't even <mask> up.
H: The police don't show up <mask> nobody <mask> hurt.
l: entailment
Perturbed: P: It's kind of weird here the way we go uh here if you have an accident and no one's hurt the police won't even get up.
H: The police don't show up even nobody gets hurt.
l: entailment

Original: P: One of our number will carry out your instructions minutely.
H: A member of our team will execute your orders with great precision.
l: entailment

Biased feature: $x = $ carry    $\hat{p}(l|x) > \frac{1}{3}$
Masking rule: Mask carry, randomly mask other words.
Masked: P: One of our number will <mask> <mask> your instructions minutely <mask>
H: A member of our team will execute your <mask> with <mask> <mask>.
l: entailment
Perturbed: P: One of our number will execute all your instructions minutely correct.
H: A member of our team will execute your instructions with immense efficiency.
l: entailment

**hypothesis-only prediction**

Original: P: Muller and most of his staff can be expected not to cause any more of the usual mid-night disturbances. .
H: Muller will most likely cause more trouble.
l: contradiction

Biased feature: $x = $ prediction of a hypothesis-only model is entailment    $\hat{p}(l|x) < \frac{1}{3}$
Masking rule: Preserve H, randomly mask P.
Masked: P: Muller and most of <mask> staff can be expected <mask> to cause <mask> more <mask> the usual pay-night disturbances.
H: Muller will most likely cause more trouble.
l: contradiction
Perturbed: P: Muller and most of his staff can be expected her to cause much more than the usual pay-night disturbances.
H: Muller will most likely cause more trouble.
l: contradiction

Original: P: My walkman broke so i'm upset now i just have to turn the stereo up real loud.
H: I'm upset because my walkman broken and now I have to turn the stereo up really loud.
l: entailment

Biased feature: $x = $ prediction of a hypothesis-only model is entailment    $\hat{p}(l|x) > \frac{1}{3}$
Masking rule: Preserve P, randomly mask H.
Masked: P: My walkman broke so i'm upset now I just have to turn my stereo up real loud.
H: I'm <mask> <mask> my walkman broken and now I have to turn my stereo up really <mask>.
l: entailment
Perturbed: P: My walkman broke so i'm upset now I just have to turn my stereo up real loud.
H: I'm sorry that my walkman broken and now I have to turn my stereo up really high.
l: entailment

**lexical overlap**

Original: P: She had come to know Betty Currie very well.
H: She had come to quite dislike Betty Currie.
l: contradiction

Biased feature: $x = $ word overlap between H and P > 0.8    $\hat{p}(l|x) < \frac{1}{3}$
Masking rule: Randomly mask unique words in P and H.
Perturbed: P: She had come to <mask> Betty Currie very well.
H: She had come to to <mask> dislikeBetty Currie.
l: contradiction
Perturbed: P: She had come to meet Betty Currie very well.
H: She had come to really dislike Betty Currie.
l: contradiction

Original: P: They were pushing the pace all right.
H: They were pushing the pace.
l: entailment

Biased feature: $x = $ word overlap between H and P > 0.8    $\hat{p}(l|x) > \frac{1}{3}$
Masking rule: Randomly mask words shared by P and H.
Masked: P: They were <mask> the pace all right.
H: They <mask> pushing the <mask>.
l: entailment
Perturbed: P: They were cleaning the pace all right.
H: They are pushing the car.
l: entailment

Table 14: A demonstration of how we conduct our perturbations under different situations for NLI.