# OpenReview forum: "Debias NLU Datasets via Training-free Perturbations"
_EMNLP/2023/Conference — EMNLP 2023 Findings_

### Official Review · Reviewer_syBo · 2023-08-06

**Soundness:** 4

**Excitement:**

3: Ambivalent: It has merits (e.g., it reports state-of-the-art results, the idea is nice), but there are key weaknesses (e.g., it describes incremental work), and it can significantly benefit from another round of revision. However, I won't object to accepting it if my co-reviewers champion it.

**Paper Topic And Main Contributions:**

This paper proposes a method for training models that are robust to spurious correlations. The method works by conducting perturbations on biased examples. The authors evaluate their proposed method on MNLI and FEVER.

**Post rebuttal changes**
1. Increased Soundness score (+1)
2. Increased Excitement score (+1)

**Questions For The Authors:**

- Given that you mention throughout the paper that the method is cost-effective, have you considered demonstrating that quantitatively?
- Have you considered training the model on e.g. MNLI and then evaluating transfer performance on multiple out-of-domain test sets? (see: Rabeeh Karimi Mahabadi, Yonatan Belinkov, and James Henderson. 2020. End-to-end bias mitigation by modelling biases in corpora. In Proceedings of the 58th Annual Meeting of the Association for Computational Linguistics.)

**Reasons To Accept:**

- Paper is well written and easy to follow.

**Reasons To Reject:**

- Novelty of the method is limited. Overall, this paper feels like a trivial extension of prior work (Wu et al., 2022).
- No experiments on QQP/PAWS and performance improvements are marginal at best, and the proposed method is considerably more complex compared to prior methods that rely on simple example re-weighting (e.g. Mahabadi et al., 2020).
- No performance comparisons with methods that do not assume prior knowledge about spurious correlations.
- Analysis/Ablations are far from comprehensive.
- NLU is a very broad term, given the datasets you are using for examples it would be more suitable to change the title and text from NLU to NLI (or alternatively, consider conducting additional experiments on tasks such as QA).

**Reproducibility:**

4: Could mostly reproduce the results, but there may be some variation because of sample variance or minor variations in their interpretation of the protocol or method.

**Reviewer Confidence:**

5: Positive that my evaluation is correct. I read the paper very carefully and I am very familiar with related work.

---

> ### Author Rebuttal · Authors · 2023-08-28
>
> We humbly believe that we have addressed your concerns below, and hope that you will consider increasing the evaluation score.
> - comments1: About novelty \
> Thank you for raising the concern regarding the novelty of our study. We appreciate your attention to previously published studies in the field. We would like to address your comment and explain why our study still contributes to the existing knowledge and advances the field.\
> While it is true that the studies by Wu et al. have adopted the framework of z-statistics, they still follow the regular way in which they finetune a generator and then filter generated samples. The finetuning-then-filtering pattern is not a perfect fit for the z-statistics framework. The z-statistics only works in the filtration process and is agnostic to the generation process. \
> However, in PDD, we attempt to further explore the potential of the z-statistic and design the whole algorithm around the z-statistic framework. We combine the advantages of PLMs and z-statistics and successfully achieve the goal of debiasing datasets while saving the trouble of training generators. Additionally, in spite of a similar definition of debiased datasets, the generation strategy and the details are quite different between PDD and Z-aug. \
> Therefore, we humbly believe that PDD is a not trivial extension but a meaningful attempt to explore the potential of z-statistics in debiasing NLU datasets.
> - comments2: Experiments on QQP/PAWS\
> Following your suggestion, we conduct the experiments on QQP/PAWs. We select "word overlap < 0.8" and "unigrams" as task-independent features. Experimental results show that PDD also helps to improve model's performance on PAWS (42.3%) and achieve a gain of 6.5% against the BERT baseline (35.8%). The performance is comparable with Regularized-conf (43.0%) but is inferior to Reweighting(48%) and PoE (55.7%). Combined with PoE, the accuracy reaches 57.9%.
> - comments3: Comparison with methods that do not assume prior knowledge about spurious correlations.\
> We thank the reviewer for pointing out this issue. We will compare PDD with several SOTA automatic debiasing methods in the revised version.
> - comments4: Incomprehensive ablations and analysis\
> We thank the reviewer for providing constructive suggestions and helping us further enrich the analysis. We will enrich the ablation experiments to cover more components of PDD in the revised version.
> - comments5: NLU to NLI \
> Thanks for the suggestion. We check the effectiveness of our framework in both NLI and Fact verification. And we conduct additional experiments on QQP/PAWS, so we prefer to remain "NLU".
> - Q3_A: Demonstrating cost quantitatively \
> Compared to Z-aug (Wu et al., 2022) which reports 24 hours spent on training the generator and 35 hours spent on generating samples with a single A100, we roughly spend 30 hours to debias each dataset with a single RTX 3090. It takes about the same amount of time to generate the data and we save time training the generator.
> - Q3_B:  Transfer performance\
> Following your suggestion, we evaluate the transfer performance of PDD on several datasets including MPE, MRPC, QNLI, WNLI, SCITAIL, SICK, and RTE. We find that the BERT-base trained on debiased MNLI totally get a 0.87% gain on average and performs better than the baseline on 6 test sets (MPE, MRPC, QNLI, WNLI, SCITAIL, and RTE). We will add this part to our analysis section.

---

### Official Review · Reviewer_gCCW · 2023-08-09

**Soundness:** 3

**Excitement:**

3: Ambivalent: It has merits (e.g., it reports state-of-the-art results, the idea is nice), but there are key weaknesses (e.g., it describes incremental work), and it can significantly benefit from another round of revision. However, I won't object to accepting it if my co-reviewers champion it.

**Paper Topic And Main Contributions:**

Existing data-centric approaches for debiasing models are still faced with the limitation of high cost, this paper proposes a framework that conducts training-free perturbations on samples containing biased features to debias NLU datasets. This framework is cost-effective, and gets improvement in several OOD test sets.

**Reasons To Accept:**

This paper proposes a framework for data-centric debiasing methods with low cost, and this method obtain gains in 2 tasks.

**Reasons To Reject:**

1. The performance improvement is weak, and is not novel.
2. The paper should conduct experiments on QQP and the corresponding PAWS dataset.
3. The ablations and analysis should be more comprehensive.

**Reproducibility:**

4: Could mostly reproduce the results, but there may be some variation because of sample variance or minor variations in their interpretation of the protocol or method.

**Reviewer Confidence:**

4: Quite sure. I tried to check the important points carefully. It's unlikely, though conceivable, that I missed something that should affect my ratings.

---

> ### Author Rebuttal · Authors · 2023-08-28
>
> We appreciate your acknowledging our framework as effective, easy-to-reproduce, and cost-effective. We will address your concerns below, and we humbly hope that you will consider increasing the evaluation score.
> - About improvement \
> The results show that PDD has achieved comparable performances with SOTA methods . Additionally, PDD is model-agnostic and can further benefit from other model-centric methods.
> - About novelty \
> Thank you for raising the concern regarding the novelty of our study. We appreciate your attention to previously published studies in the field. We would like to address your comment and explain why our study still contributes to the existing knowledge and advances the field.
> While it is true that the studies by Wu et al. have adopted the framework of z-statistics, they still follow the regular way in which they finetune a generator and then filter generated samples. The finetuning-then-filtering pattern is not a perfect fit for the z-statistics framework. The z-statistics only works in the filtration process and is agnostic to the generation process.
> However, in PDD, we attempt to further explore the potential of the z-statistic and design the whole algorithm around the z-statistic framework. We combine the advantages of PLMs and z-statistics and successfully achieve the goal of debiasing datasets while saving the trouble of training generators. Additionally, in spite of a similar definition of debiased datasets, the generation strategy and the details are quite different between PDD and Z-aug.
> Therefore, we humbly believe that PDD is a not trivial extension but a meaningful attempt to explore the potential of z-statistics in debiasing NLU datasets.
> - Experiments on QQP/PAWS\
> Following your suggestion, we conduct the experiments on QQP/PAWs. We select "word overlap < 0.8" and "unigrams" as task-independent features and find PDD also helps to improve model's performance on PAWS (42.3%) and achieve a gain of 6.5% against the BERT baseline (35.8%). The performance is comparable with Regularized-conf (43.0%) but is inferior to Reweighting(48%) and PoE (55.7%). Combined with PoE, the accuracy reaches 57.9%.
> - Complexity of PDD\
> Notice that those sample-reweighting methods are model-level and orthogonal to our data-level framework. So it is not appropriate to directly compare them. Compared with existing data-level methods such as Z-aug, AFLITE, TAILOR and so on, PDD exhibits no more complexity.
> - Incomprehensive ablations and analysis\
> We thank all the reviewers for providing constructive suggestions and helping us further enrich the analysis. We will enrich the ablation experiments to cover more components of PDD in the revised version.

---

### Official Review · Reviewer_TmsF · 2023-08-11

**Soundness:** 3

**Excitement:**

4: Strong: This paper deepens the understanding of some phenomenon or lowers the barriers to an existing research direction.

**Paper Topic And Main Contributions:**

The paper deals with the problem of bias in NLU datasets, specifically in the form of spurious correlations between the features and labels that models tend to latch on for their prediction, but that fail to generalize to out-of-distribution (OOD) situations, therefore causing a performance drop. To address this issue the paper proposes a method to debias training datasets by augmenting them with new generated samples crafted so as to remove the spurious correlations that drive the bias are removed. The new samples are generated via a perturbation of the original training samples obtained through a pretrained mask language model and task-specific prompt templates in order to enforce a conditioning on the label when generating the new sample.
The method is evaluated on two NLU tasks, NLI and Fact Verification, and is shown to outperform state-of-the-art debiasing methods on both in-domain and out-of-domain datasets.

**Post-rebuttal period changes:**
* Increased "Excitement" to "strong"
* Increased "Confidence" to "pretty sure"

**Questions For The Authors:**

- Is there any sense that the method could be generalized to enable automatic discovery of the spurious correlations and the features that are responsible for the bias and OOD performance drop?
- PDD is applied only so long as the z-values of the biasing features are not uniform. On the other hand, one might imagine running the algorithm for longer to generate even more perturbed samples (making sure that the z-values remain uniform), thereby using it as a data augmentation procedure. Would that help improving performance even further or does the performance improvement plateaus very quickly as soon as the z-values are uniform?

**Reasons To Accept:**

- The debiasing method proposed in the paper is duly benchmarked against multiple state-of-the-art debiasing strategies
- The proposed debiasing method is a data-centric approach that can be combined with any available model-centric debiasing method to take advantage of the combined capabilities
- The proposed debiasing method matches or beats the performance of competitor methods on both ID and OOD datasets

**Reasons To Reject:**

- The work is limited to NLI and Fact verification tasks, the biases that this specific type of task displays, and the features that capture them
- The method is highly reliant on domain knowledge to identify the features onto which the spurious correlations latch to drive the bias

**Reproducibility:**

4: Could mostly reproduce the results, but there may be some variation because of sample variance or minor variations in their interpretation of the protocol or method.

**Reviewer Confidence:**

3: Pretty sure, but there's a chance I missed something. Although I have a good feel for this area in general, I did not carefully check the paper's details, e.g., the math, experimental design, or novelty.

---

> ### Author Rebuttal · Authors · 2023-08-28
>
> - Comments & Q1_A: "Limitation and reliance on prior knowledge" \
> According to the framework of PDD, the limitation and the heavy reliance on prior knowledge are attributed to the manually-designed task-independent features. An automatic way is using a model that is able to capture various spurious correlations. For example, we can use the prediction of a shallow model trained on a small subset of training sets as the feature (Prasetya Ajie Utama, Nafise Sadat Moosavi, and Iryna 686 Gurevych. 2020b. Towards debiasing NLU models 687 from unknown biases. In EMNLP 2020.) and then debias the dataset.  We will explore its feasibility in future work.
> - Q1_B:"The potential to serve as a data augmentation procedure" \
> Since the threshold of z-values is adjustable, we can set threshold = 0 to make sure the perturbation procedure will never stop in order to generate more samples while keeping the z-values uniform. We conduct the experiment on the debiased MNLI (590K) and produce a larger dataset containing 740K samples but find no obvious performance improvement. And we will try to investigate the influence of data size on OOD performance in our analysis part.

---

### Meta-Review · Area_Chair_acmz · 2023-09-18

**Recommendation:** 4

**Metareview:**

This paper introduces a novel data-augmentation method to reduce spurious correlations when training on NLU datasets. Proposed method uses another pre-trained Masked-LM to generate perturbations input features and therefore reduces the cost compared to previous work (Z-aug). Reviewers agree on the soundness of the results and find them interesting.

---

### Decision · Program_Chairs · 2023-10-07

**Decision:**

Accept-Findings

**Comment:**

This paper introduces a novel data-augmentation method to reduce spurious correlations when training on NLU datasets. Proposed method uses another pre-trained Masked-LM to generate perturbations input features and therefore reduces the cost compared to previous work (Z-aug). Reviewers agree on the soundness of the results and find them interesting.